# Common Determinants of Dental Caries and Obesity in Children: A Multi-Ethnic Nested Birth Cohort Study in the United Kingdom

**DOI:** 10.3390/ijerph182312561

**Published:** 2021-11-29

**Authors:** Magdalena F. Uerlich, Sarah R. Baker, Peter F. Day, Lucy Brown, Mario V. Vettore

**Affiliations:** 1Unit of Oral Health: Dentistry and Society, School of Clinical Dentistry, University of Sheffield, Sheffield S10 2TN, UK; mfu@uerlich.net; 2Department of Paediatric Dentistry, School of Dentistry, University of Leeds, Leeds LS2 9JT, UK; p.f.day@leeds.ac.uk; 3Bradford Community Dental Service, Bradford District Care NHS Foundation Trust, Bradford BD7 3EG, UK; 4Harrogate and District Foundation Trust, Community Dental Services, Harrogate HG2 7SA, UK; lucy.brown14@nhs.net; 5Department of Health and Nursing Science, University of Agder, 4630 Kristiansand, Norway; mario.vettore@uia.no

**Keywords:** dental caries, paediatric obesity, structural equation modelling, children’s oral health, common risk factor

## Abstract

The article examines the common determinants of childhood dental caries and obesity. Longitudinal data from the Born in Bradford cohort study (BiB1000) (n = 1735) and dental data (dental general anaesthetics (GA) and oral health survey 2014/15) (n = 171) were used to test a framework on the social determinants of childhood dental caries (decayed, missing, filled teeth (dmft) index) and obesity (body mass index (BMI)). The BiB1000 data were collected at pregnancy week 26–28 and after birth at 6, 12, 18, 24 and 36 months. The determinants were demographics, wellbeing, socio-economic status (SES), dietary behaviours and physical activity behaviour of the children. Missing data were accounted for through multiple imputation (MI). The framework was tested through structural equation modelling. Overall, the model fit was adequate. No alcohol consumption of the mother after giving birth, higher frequency of child drinking sugar-sweetened beverages, emotional and behavioural difficulties of the child and being male were directly associated with both BMI and dental caries. Caregivers uninvolved or indulgent feeding style were associated with higher BMI and less dental caries. Social deprivation was associated with lower BMI and higher dmft. Five determinants were directly associated with BMI only. Fifteen indirect paths were significant for both child dental caries and BMI. The findings suggest common determinants for both childhood obesity and dental caries. Common risk factor approach seems appropriate for planning future health promotion programmes.

## 1. Introduction

Childhood obesity and dental caries are two of the foremost public health challenges of the 21st century. In recent years, the prevalence of childhood obesity has significantly increased among developing and developed countries alike [1]. Recent findings of the UK Government’s National Child Measurement Programme reported that more than one third (34.3%) of children in year six (aged 10/11 years old) were considered either overweight or obese in England [2].

Overweight and obesity is the imbalance between energy intake and energy expenditure [1]. Such imbalance is often associated with multiple factors, such as inadequate diet and lack of physical activity that are influenced by environmental and societal characteristics [1]. Overweight/obesity during childhood are also associated with chronic diseases in later life, including diabetes, cardiovascular diseases, musculoskeletal disorders and some types of cancers [1]. Obese children have an increased risk for hypertension, insulin resistance, fractures and psychological problems [1]. Overweight/obesity are also associated with adverse social impacts such as discrimination, bullying and social exclusion during childhood [3] as well as a higher chance of obesity, premature death and different types of disabilities in adulthood [1].

More than 530 million children worldwide suffer from dental caries in primary teeth [4]. In the UK, in 2019, tooth extraction was the main reason for hospital admission among children aged between six and ten years [5]. Even though the prevalence of decayed teeth in children in the UK has significantly decreased in recent years, oral health inequalities between socio-economic groups persist [6]. Dental caries may cause toothache [7], tooth loss and systemic infection [4]. In addition, it may influence children’s school performance and missing school days [4,8] and may result in difficulties in eating and sleeping due to pain [9].

Recent systematic reviews have identified a sound amount of published work on the possible link between childhood dental caries and obesity [10,11,12]. However, there was a clear inconsistency across the findings of the studies possibly due to methodological flaws and analytical issues, including dental caries and obesity measurements, lack of control for potential confounders and prevalence bias since most studies were cross-sectional. The discrepancies of previous results may also be related to social background and cultural differences of the studied samples. The main gaps in the literature include the lack of longitudinal data, the absence of a theoretical framework to guide the selection of the variables and the adoption of a robust statistical approach to account for the above-mentioned issues. To date, few studies have investigated the common determinants of dental caries and obesity. The common risk factors approach identifies a number of risk factors that can increase the risk for multiple diseases. Therefore, tackling common risk factors can lead to the reduction of several diseases [13]. Enhancing the understanding of the possible common determinants of dental caries and obesity may result in more effective interventions and strategies to tackle these diseases according to the common risk factor approach [14].

A theoretical framework [15] was developed to investigate the common determinants of childhood dental caries and obesity based on the ecological model of predictors of childhood overweight [16] and the social determinants of oral health [17] (Figure 1).

The determinants of childhood obesity and dental caries were organized into family- and childhood-level. According to the proposed framework, socio-economic status of the family would predict parental health beliefs, which in turn would influence health behaviours and practices of the parents as well as their coping skills. For instance, living in an underprivileged family would potentially lead parents to adopt unhealthy dietary habits because they think that having a healthy diet is not important for their health (negative health beliefs). It was anticipated that family-level factors would be directly related to child health behaviours and practices. This means that a child’s routine health behaviours, such as pattern of physical activity, are shaped by their parent’s level of physical activity. Child health behaviours and practices would also be predicted by child physical and demographic characteristics. Ultimately, dental caries and overweight/obesity would be directly influenced by physical and demographic characteristics of the child as well as child health behaviours and practices. Moreover, the latter would mediate the link of family-level determinants with dental caries and overweight/obesity in children. The aim of this study was to examine the common determinants of childhood overweight/obesity and dental caries according to the proposed framework.

## 2. Materials and Methods

### 2.1. Data Sources

This study combined data from three separate sources: (i) the Born in Bradford Cohort study and dental data from (ii) general anaesthetics (GA) episodes and (iii) the Oral Health Survey of 5-year-old Children 2014–2015. These data sources were selected based on the following aspects. Data were obtained through valid and reliable methodological procedures, the similar age of participants among the datasets and the availability of relevant variables to test the proposed adapted framework (Figure 1). Ethical approval for the BiB study was granted by Bradford Research Ethics Committee (Ref 07/H1302/112) and all women who agreed to participate signed a consent letter before data collection. Children who have participated in the Born in Bradford Cohort Study and had undergone dental care under GA were asked for data linkage (16/YH/0048). The elective dental care provided under GA is provided by one organisation in Bradford namely the Salaried Dental Service, Bradford District Care Trust. Parents gave consent for data linkage prior to this study and data were linked based on NHS-number following ethical approval (16/YH/0047 and 16/YH/0048).

#### 2.1.1. The Born in Bradford Cohort Study

The Born in Bradford Cohort (BiB) Study is a multi-ethnic longitudinal birth cohort study that collected information on health and wellbeing of over 13,500 children and their parents residing in Bradford, UK [18]. The study recruited women who were between the 26th and 28th week of gestational age and their children who were born between 2007 and 2011 at the Bradford Royal Infirmary. Mothers, children and their families have been followed from pregnancy through the childhood to investigate the social and environmental influences on health and diseases. The BiB data included: (i) mother’s baseline questionnaire conducted between the 26th and 28th week of gestational age and (ii) the BiB1000 cohort, a nested cohort within the wider BiB cohort [19]. Of 1916 eligible women, 1735 agreed to participate in the BiB1000 cohort. Of them, 28 mothers gave birth to twins and were therefore excluded as the statistics provided here were for single births only (n = 1707). Data were collected at the 6 (77% of the BiB1000 cohort participated), 12 (75%), 18 (74%), 24 (70%) and 36 (70%) months of child’s age. The Oral Health Survey of 5-year-old Children 2014-2015 is part of the Public Health England Dental Public Health Epidemiology Programme and is coordinated and carried out by Public Health England [20]. Dental data were collected using visual method by trained and calibrated examiners who were generally employed by NHS trusts providing community dental services [20]. Parental consent was obtained prior to the data collection [20]. Further information on the data collection procedures and equipment can be found in the primary publications [18,19,20,21].

#### 2.1.2. Dental Data

Dental data of the children participating in BiB study were derived from two sources: dental GA episodes (first sequence of dental GA, n = 1071) [21] and the Oral Health Survey of 5-year-old Children 2014–2015 (n = 316) [20]. Dental GA data were derived from records of all children’s routine hospital admissions between 2009–2016 (≈8000) for dental care under general aesthetic in Bradford.

The Oral Health Survey of 5-year-old Children was conducted in the school year 2014/2015 to estimate the prevalence and severity of dental caries among 5-year-old children within each lower tier local authority [20]. The Oral Health Survey of 5-year-olds was conducted at mainstream primary schools in England and included 111,500 children [20]. Demographics (NHS number, full name, date of birth and address) of the children participating in the oral health survey were registered before undergoing a short dental examination to collect dmft index. Ethical approval was granted by Public Health England and parental consent were obtained prior to the data collection [20]. Of the 324 children who participated in the BiB Study and in the Oral Health Survey of 5-year-old Children 2014–2015, 316 provided consent for data linkage. Participants’ data of the BiB study and the oral health survey were anonymously linked using encrypted NHS numbers [21].

### 2.2. Outcome Measures

#### 2.2.1. Nutritional Status

Nutritional status was assessed using the body mass index (BMI) to estimate the proportion of children with overweight/obesity. BMI is calculated using the following formula: BMI = weight (in kg)/height² (in m). BMI classification in children also considers the cut off scores for overweight/obesity according to sex and age, referred as BMI for age, percentiles or z-scores. The z-score is calculated as follows: z-score (or SD-score) = (observed value—median value of the reference population)/standard deviation value of reference population. The WHO reference charts for children were used to calculate the BMI, not differentiating between ethnicities [22].

In this study, BMI data were calculated using the BiB1000 dataset. Since not all children participated in all appointments of growth measurement throughout the years in the BiB study, BMI data were taken from one point in time using the growth measurement closest to the dental GA appointment or closest to the date of the oral health survey examination.

#### 2.2.2. Dental Caries

The dmft index was employed to assess experience of dental caries in primary teeth, according to the following codes: ”0 = sound tooth”, “1 = extracted tooth”, “2 = filled tooth”, “3 = fissure sealed tooth” and “4 = crowned tooth”. The data were recoded as dental caries (original codes 1, 2 and 4) and sound teeth (original codes 0 and 3) in order to be comparable with the dmft data from the oral health survey. The dmft scores were obtained by summing the number of extracted, filled and crowned teeth.

#### 2.2.3. Family-Level and Child-Level Determinants

The included determinants of dental caries and overweight/obesity were chosen based on the Fisher-Owens [17] and Birch and Davison [16] theoretical frameworks, respectively, and can be found in Table 1.

Mother’s baseline data derived from the BiB1000 dataset were collected at the 26th–28th week of gestation and included information concerning family socio-economic status, maternal depression, children’s wellbeing, growth and obesity patterns, childhood physical activity and children’s diet.

Some original variables of the BiB1000 study were combined to create the determinants of the proposed frameworks. For example, fruit and vegetable consumption, fatty food consumption and amount of consumed food were used to generate the variable diet. The BiB1000 data were collected at 6, 12, 18, 24 and 36 months of child’s age and the variables were chosen from all follow-up periods due to the variation on data availability across the follow-ups. Thus, the proposed framework was populated through combining data from the BiB1000 waves and from the wider BiB cohort study (SES, gender, ethnicity, weight and height and mother’s age at birth). Data of determinants were chosen considering the closest date of the dental data examination. Due to data availability, children were between 0 and 5 years old at the time of anthropometric measurement with a mean age of 3.88 years.

### 2.3. Data Analysis

#### 2.3.1. Descriptive Analysis

The demographic characteristics, nutritional status and dental caries were described using means, standard deviations and range for continuous data and frequencies for categorical variables according to the three datasets (BiB study, Dental GA and Oral Health Survey).

#### 2.3.2. Missing Data and Multiple Imputation

Missing data from the BiB1000 dataset were replaced by data from the overall BiB study. They included data on children’s demographics (e.g., sex), weight and height measurements and social deprivation. Of the 15 variables included in the analysis, six had complete data for all participants (n = 171). Missing data of the remaining nine variables ranged from 18% to 36% of the dataset (Appendix A, Table A1). Of them, 8 variables were included in final analysis (parsimonious model).

Multiple imputation (MI) method was used to deal with missing data through generating and combining several different, plausible imputed datasets [27]. The original data are used to estimate multiple values that reflect the uncertainty around the true value. The Markov Chain Monte Carlo procedure was adopted to impute multiple variables at the same time. This method is considered adequate in social research since it assumes that all the variables of the imputation model have a joint multivariate normal distribution (mvn) [28]. The data augmentation algorithm (mnv) is used to replace the missing data by drawing from the conditional distribution and most theoretically sound if the sample is large enough. Therefore, the mvn is a sound method even when the normality assumption is violated [28]. Most likely, when using the mvn method, the imputed data for categorical variables will be greater or smaller than some of the original categories within the variable [28].

In the first stage of the MI (imputation phase), data are imputed, and missing data is replaced by the dataset. The analysis phase is the second MI stage, where each dataset is analysed with the chosen statistical method. Finally, the estimated parameters of each dataset are combined in the pooling phase for inference. Further details are available on Appendix B.

#### 2.3.3. Structural Equation Modelling

The direct and indirect relationships of family-level and child-level determinants with nutritional status and dental caries were tested using Structural Equation Modelling (SEM). The full model was initially tested for identification. Then, a statistically parsimonious model was estimated after the removal of the non-significant direct paths. The direct and indirect standardized effects and 95% confidence intervals (CI) were obtained through MI to account for the missing data. Multivariate normal distribution (mvn) method was used to impute multiple datasets that were linked with SEM thereafter [29]. Fit indices of the SEM models could not be calculated once the MI procedure generates multiple models/datasets [30]. All analyses were conducted using STATA software version 22.0.

## 3. Results

The studied sample included 171 children with dental data who participated in the BiB1000 nested cohort, including dental GA data (n = 136, mean age at the time of operation was 5.7 years, range: 2.3–6.6 years) and Oral Health Survey of 5-year-old Children 2014–2015 (n = 35, mean age at the time of examination was 5.4 years, range: 4–6 years).

In total 23.4% of the children were overweight or obese. The mean of dmft index among participants of the GA sample was 9.1. Of the 35 children participating in the Oral Health Survey of 5-year-old children 2014–2015, 23 were caries-free. Of the 12 with decay the average dmft was 2.8. Overall, 46.2% of the studied sample was composed of male children (Table 2).

More than half of the children (57.4%) were from families receiving benefits or deprived families. In addition, the majority of the sample was of Pakistani origin (60.2%). Appendix C, Table A2 presents the number of participants and percentages of the determinants of the family- and child-level characteristics. Both were calculated using the original data, before MI was conducted and therefore include a maximum of 171 children without missing data from the BiB1000 dataset.

Six family- and child-level determinants were directly linked with both BMI and dental caries (Figure 2).

No alcohol consumption of the mother after giving birth, higher frequency of child drinking sugar-sweetened beverages, emotional and behavioural difficulties of the child and being male were directly associated with higher BMI and more dental caries. In addition, caregivers uninvolved or indulgent feeding style were associated with higher BMI and less dental caries. Social deprivation was directly related to lower BMI and higher number of carious teeth. The direct effects and 95% CI are presented as Appendix D, Table A3.

Higher maternal age at birth and absence of breastfeeding were directly associated with lower child BMI. Parental psychological distress and Pakistani ethnicity or other ethnicities were linked with a higher BMI of the child. Greater frequency of child physical activities was directly associated with higher BMI (Figure 2 and Appendix D, Table A3).

Significant indirect relationships of family- and child-level determinants with BMI and dental caries were identified. Dental caries and BMI were indirectly related to mother’s age at birth, deprivation status, caregivers feeding style, parental psychological wellbeing, no alcohol consumption mother, breastfed, physical activity of the child, ethnicity and daily TV hours during weekday of the child (Appendix E).

## 4. Discussion

Overall, both family- and child-level variables were relevant determinants of dental caries and overweight/obesity. Of the 15 hypothesised determinants of dental caries and BMI, 6 were linked with dental caries and overweight/obesity, including lack of alcohol consumption of the mother after giving birth, high frequency of drinking sugar-sweetened beverages of the child, lower behavioural and emotional wellbeing of the child and being male were significant determinants of overweight/obesity and dental caries. Higher social deprivation also directly influenced lower BMI and higher dental caries. An uninvolved feeding style was linked to higher BMI and a lower dental caries. Further five determinants were only associated with BMI, but not with dental caries; mother’s age at birth, parental psychological wellbeing, breastfeeding, child’s physical activity and ethnicity.

The significant association of higher frequency of sugar sweetened beverages consumption with dental caries and overweight/obesity was a confirmatory finding, since a number of previous studies reported such relationship [31,32]. The second significant common determinant of both conditions was sex. In this study, being male was related to childhood overweight/obesity and dental caries. Sex differences in the BiB population may be due to cultural differences because of the high percentage of Pakistani children in the sample. In this population, boys often have more access to sugary and rich calorie drinks and diets than girls. Sex differences in childhood overweight/obesity and dental caries have been reported elsewhere [33,34]. One possible explanation for this finding is that boys often have more access to sweets and calorie rich foods than girls in some cultures [34]. The third significant determinant was child’s emotional and behavioural state. Children with greater emotional and behavioural difficulties were more likely to experience overweight or obesity and dental caries. However, it is not possible from the BiB data to detail what emotional and behavioural difficulties children were experiencing to understand more about how this relates to obesity or tooth decay. The generalization of our findings to general population must be cautious since most of participants (57.4%) were from deprived families or from families receiving benefits. In addition, around 60% of the participants were of Pakistani origin.

In addition to the child-level determinants, three family-level determinants were found to be significant for both dental caries and overweight/obesity. They were social deprivation, caregivers feeding style and lack of maternal alcohol consumption. Children who grew up in social deprivation were more likely to have more dental caries. Otherwise, social deprivation was inversely associated with overweight or obese children. Previous research has reported the link between dental caries and higher levels of social deprivation [6]. However, the reported association between low social deprivation and overweight/obesity are not in accordance with other studies. A higher social deprivation status has been commonly linked to overweight/obesity. Though, ethnic background of the studied sample may account for the difference between our findings and previous studies [35]. The majority of Pakistani people living in the UK belong to the lowest two quintiles of income [36] and previous research suggests that Pakistani children living in the UK are 40% less likely to be obese than white children [36].

Uninvolved caregiver’s feeding style was associated with childhood overweight/obesity. Moreover, authoritarian feeding style was linked to childhood dental caries. Uninvolved feeding styles have been found to lead to unhealthy eating behaviours in children and authoritarian feeding styles to resistance of children and fussiness in terms of food preferences [25,37]. These findings suggest that both extremes of caregivers feeding style might lead to childhood dental caries and overweight/obesity. The lack of alcohol consumption during the postpartum period was associated with dental caries and overweight or obesity in children in this study. These findings might be considered surprising and specific to the study population due to the high percentage of Pakistani participants. The Pakistani ethnic group living in England is composed of the highest percentage of non-drinkers [38]. However, alcohol consumption in the BiB study was only measured once, after giving birth, and the amount of alcohol consumption was not assessed.

One of the key strengths of the present study was the use of a comprehensive conceptual framework to select the determinants and to guide the SEM analysis to test the hypothesised relationships. Even though conceptual framework supported the understanding of the determinants of childhood overweight/obesity and dental caries, the framework did not consider the community- and national-level determinants of both health outcomes. The assessment of those characteristics in future research will enhance the understanding of the broader determinants of childhood overweight/obesity and dental caries. The use of secondary data (BiB data set and dental data) allowed the researcher to test the conceptual framework using SEM. SEM is considered a robust statistical method to test complex relationships between determinants that can be simultaneously assessed as independent and dependent variables following a theoretical model.

Data linkage was necessary to combine fifteen relevant family and child-level determinants for dental caries and BMI into a single dataset according to the proposed framework data. However, the dataset linkage process resulted in missing data and reduced the expected sample size. Consequently, the precision and the power of the study was lower than initially planned. Missing data was due to the unavailability of dental data for some of the participants of the BiB study. In addition, participants without dental data were excluded for the purpose of this study. Some participants with dental data had little BiB1000 data available and therefore were also excluded. This was mainly due to the fact that BiB1000 data originated from a longitudinal study with different time points of data collection and some participants did not participate in data collection for each of the six points in time. Our findings might have been affected by the fact that some family and child-level determinants were only collected at once and in different time points. For instance, maternal depression at 18 months after birth. Data collection at an earlier or later point in time might have affected the results.

MI was performed in order to account for the missing data, allowing the researcher to conduct the analysis despite the missing data. MI was chosen over the more commonly used ML estimation as the latter method is only suitable for data that are normally distributed and therefore most likely continuous data. However, most of the variables in this dataset was categorical. Another strength of using the MI method is obtaining unbiased estimates and more accurate results. As a result, the findings are more likely to represent the nature of the events. This occurs through the estimation of missing data, by combining the results of multiple imputed datasets and taking the average of these results as new values to replace missing data. Nevertheless, even though some of the missing data were accounted for, it must be acknowledged that using primary data would have been preferable rather than estimating missing data. This is because the former would result in more accurate measurements of the characteristics of the study population.

Common family and child-level determinants of childhood dental caries and overweight/obesity were identified. Combining strategies to tackle multiple health outcomes is not only cost-effective for governments, but they also result in substantial health gains for the population. For example, increasing awareness on healthy eating and physical activity will affect multiple health outcomes. Future prevention campaigns and health promotion strategies to tackle childhood dental caries and overweight/obesity should target common determinants, such as emotional and behavioural wellbeing, and reduction in consumption of sugary drinks. These strategies should focus on the reduction of social inequalities that can potentially result in better population health and consequently reduce healthcare costs. The conceptual framework of this study should be tested in the future considering community- and national-level determinants. Future research in other countries, such as the United States and Australia that have implemented different health policies and programmes specifically to tackle dental caries and overweight/obesity. Additionally, qualitative studies should be carried out to explore, for example, the family and child experiences regarding food policies and their impact on school meals, grocery prices and health.

## 5. Conclusions

Four determinants—no alcohol consumption by the mother after giving birth, higher frequency of the child drinking sugar-sweetened beverages, emotional and behavioural difficulties of the child and being male—directly influenced both BMI and dental caries. Two determinants were linked to either a higher BMI and a lower dmft (caregivers uninvolved or indulgent feeding style) or a lower BMI and a higher dmft (social deprivation). Five determinants were directly associated with BMI only. Additionally, fifteen indirect paths were significant for both child dental caries and BMI. The findings suggest common determinants for both childhood obesity and dental caries.

## Figures and Tables

**Figure 1 ijerph-18-12561-f001:**
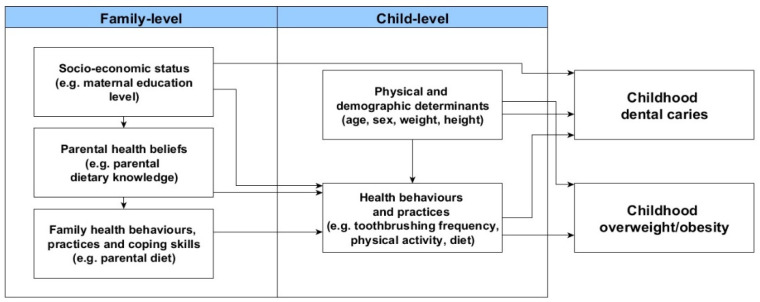
The adapted framework of the social determinants of dental caries and overweight/obesity in children [15].

**Figure 2 ijerph-18-12561-f002:**
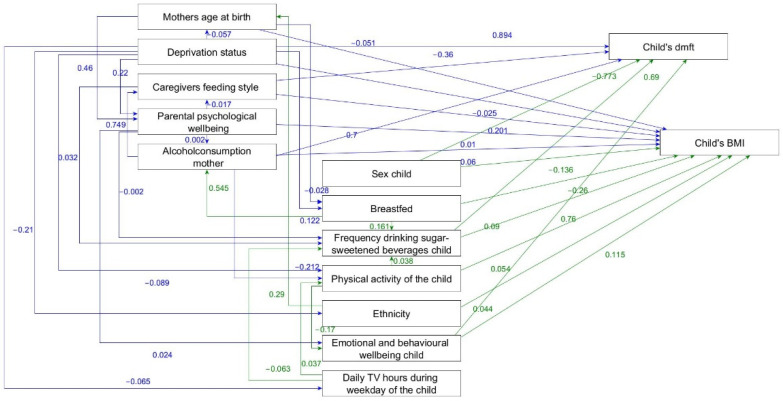
Parsimonious model of the adapted framework of the social determinants of dental caries and overweight/obesity in children indicating the direct effects non-standardised ß-coefficients.

**Table 1 ijerph-18-12561-t001:** Family and child-level determinants included in this study.

Variable Name	Source	Description of Variable	Categories
Family-Level Determinants			
Deprivation status	Various BiB sources	The variable describes the SES of the family the child grows up in. The variable derives from different variables within the BiB dataset and has been produced using latent class analysis [23]	(1) least deprived and most educated; (2) employed and not materially deprived; (3) employed but no access to money; (4) receiving benefits but coping; (5) most deprived
Mother’s alcohol consumption	BiB1000 24 months	The variable describes if the mother consumed alcohol after giving birth at any point in time and does not consider frequency or quantity.	(1) yes; (2) no
Mother’s age at birth	BiB 1000	The variable indicates the mother’s age at birth in months	(1) under 25; (2) between 25 and 29.9; (3) 30 and above
Child breastfeeding	BiB1000 6 months	The variable describes if the child was breastfed at any point after giving birth. It does not give information on the exclusivity or frequency of breastfeeding.	(1) yes; (2) no
Parental psychological wellbeing	BiB1000 18 months	The determinants assess somatic symptoms, anxiety and insomnia, social dysfunction and severe depression. This using the standardised GHQ-28 questionnaire [24].	The higher the score, the more psychological distress
Caregiver’s feeding style	BiB1000 24 months	Identifies four parental feeding styles: authoritative (high responsiveness, high demandingness), authoritarian (low responsiveness, high demandingness); indulgent (high responsiveness, low demandingness) and uninvolved (low responsiveness, low demandingness). This was measured based on the questionnaire and cut of points by Hughes and colleagues [25].	(1) authoritarian; (2) authoritative; (3) indulgent; (4) uninvolved
Child-Level Determinants			
Ethnicity	Baseline and other BiB sources	This variable describes the ethnicity of the child	(1) White British; (2) Pakistani; (3) Other
Physical activity of the child	BiB1000 24 months	The variable describes how often the child has been taken somewhere to be physically active	(1) never; (2) once a month up to once a week; (3) twice a week up to 7 times a week
Sex	Various BiB sources	Sex of the child	(1) male; (2) female
Daily TV hours during weekday of the child	BiB1000 24 months	The variable indicates the hours of TV the child watches during a 24-hr weekday	(1) none; (2) up to 1 h; (3) 1–2 h; (4) 2–3 h; (5) more than 3 h
Frequency of drinking sugar-sweetened drinks child	BiB1000 36 months	The variable describes the frequency of drinking sugar- sweetened beverages over the period of 2–3 months	(1) very low consumption of sugary drinks; (2) medium to high consumption of sugary drinks
Emotional and behavioural wellbeing of the child	BiB1000 36 months	The score of the validated strength and difficulties questionnaire [26] indicates the results of a brief emotional and behavioural screening of the child. A higher score indicates lower levels of emotional and behavioural wellbeing	(1) close to average; (2) slightly raised; (3) high; (4) very high
Hours of sleep child/day	BiB1000 24 months	The score indicates the hours a child sleeps per 24 h	(1) less than 10 h per 24 h; (2) between 10–12 h per 24 h; (3) 12.5–14 h per 24 h; (4) more than 14 h per 24 h
**Outcomes**			
Dental caries	GA dataset and oral health survey of 5-year-old children 2014–2015	The categories indicate if a child has experienced dental caries or if the teeth are caries free	Count ranging from 0–21 teeth affected by dental caries
Weight status	Various BiB sources	The variable indicates the weight status in relation to height and age of children (WHO standards)	(1) severe thinness; (2) thinness; (3) normal weight; (4) overweight; (5) obese

**Table 2 ijerph-18-12561-t002:** Demographic characteristics of child participants.

	Dental GA (n = 136)	Oral Health Survey of 5-Year-Old Children 2014–2015 (n = 35)	Total (n = 171)
Age children (years), mean (range)	5.7 (2.3–6.6)	5.4 (4–6)	5.6 (2.3–6.6)
Weight status of child, N (%)			
Severe thinness	0	0	0
Thinness	6 (4.5%)	0	6 (3.5%)
Normal weight	99 (73%)	26 (74.3%)	125 (73.1%)
Overweight	25 (18%)	8 (22.8%)	33 (19.3%)
Obese	6 (4.5%)	1 (2.9%)	7 (4.1%)
dmft: mean (SD) range	9.1 (3.9) 2–21	0.9 (1.8) 0–8	mean: 7.4
Sex, N (%)			
Male	68 (50%)	11 (31.4%)	79 (46.2%)
Female	68 (50%)	24 (65.6%)	92 (53.8%)

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
