# Peer review of "Common Determinants of Dental Caries and Obesity in Children: A Multi-Ethnic Nested Birth Cohort Study in the United Kingdom"

_ijerph, 2021, doi:10.3390/ijerph182312561_

Round 1
Reviewer 1 Report
The manuscript entitled " Common determinants of dental caries and obesity in children: A multi-ethnic nested birth cohort study in the United Kingdom” corresponds with to this journal. Manuscript is written according to the instructions for the authors. The topic is especially important due to the high incidence of caries and obesity as well as connection between caries and obesity. The results can help plan further preventive measures. References need to be corrected according to the instructions.
Author Response
Please see attachement

Reviewer 2 Report
This is very interesting paper investigating the common determinants of children’s caries and obesity in a birth cohort in the UK based on their proposed framework of common determinants involving children and family level related factors. It helps fill the gap of knowledge for the association between child caries and obesity. I have following specific comments:
- Introduction: I would suggest the authors to have a more in-depth discussion on family and child-level related factors that the authors used in their proposed framework.
- Page 3, Line 86-90: There’s inconsistency regarding the baseline time for gestational age: is that week 20 of pregnancy or week 26-28? This needs clarity.
- Some of the interesting measures such as emotional and behavioural difficulty, child breastfeeding (does ‘Yes’ measure exclusive breastfeeding or includes both exclusive breastfeeding and mixed feeding?), parental feeding style (what do you mean by different categories? how do you measure these categories?), maternal depression at baseline were not sufficiently provided with details which leaves a poor readership as readers have less information about BiB’s data collection. In addition, I wonder whether the deprivation measure is based on their address (index of deprivation measure based on postcode) or self-reported data from the questionnaire.
- Page 9, Line 255-56, could you explicitly explain what cultural difference of Pakistani affected the sex difference in this sample?
- Page 10, Line 283-85, from the abstract the finding indicates that maternal no alcohol consumption was associated with dental caries and overweight/obesity and also in your Table 3, whereas in the Discussion, it says ‘lower consumption of alcohol’, the authors need to check their presented findings throughout the paper to ensure consistency and clarity.
- Discussion: The determinants were collected at different time points respectively. For example, parental psychological wellbeing was collected at 18 m of postnatal time. Maternal psychological wellbeing may show differences at an earlier stage, for instance in the first half year of child’s age. This should be addressed as limitations of the study as the indicators collected as a specific timing might mask the change over time during the longitudinal study.
Author Response
Please see attachement

Reviewer 3 Report
Uerlich et al. analyzed longitudinal data from the Born in Bradford cohort study (BiB1000) and dental data (dental GA and oral health survey 2014/15) to test a framework on the social determinants of childhood dental caries and obesity. Overall, the manuscript is clearly written and the results are well presented. I have some concerns about the bias of data collection since 57.4% of children were from families receiving benefits or deprived families and the majority of the sample was of Pakistani origin (60.2%). Which means the results presented here might only apply to certain group. The authors should make it clear.
Detailed comments:
- Line 15, please write down the full name of dmft index, the Decayed, Missing, and Filled Teeth (DMFT) index.
Author Response
Please see attachement

Reviewer 4 Report
Dear authors,
Thank you for presenting us your work. After several readings, there were some topics that came to my attention, and I'll leave them below so that you can improve your work.
Overall, the writing is not good. It was really difficult to understand what was done and, because of that, I couldn't properly review the statistical analysis since I haven't understand what was done with which data in most of the cases.
Introduction:
- Reference 1 is repeated many times alone. Authors should add other relevant works in accordance.
- Some references appear in the middle of a sentence and no reference was added at the end, but additional information is written; please check.
- P. 2, l. 51/52 - which marked oral health inequalities persist?
- Define abbreviations at its first use (example - SES. but it happens a lot along the manuscript, please revise).
- P. 2, l. 66 - who developed this framework? Seems like it is already part of the author's work on the subject. If already published, add reference. Otherwise, this is already methods of your work, not introduction. Also, add reason for its development, whom, when,...
- P. 2, l. 68 - authors are not written on the references, which makes impossible to link it to the names at figure 1 legend.
- Overall, introduction is quite general, does not clearly specify the lacks in knowledge on the subject, and doesn't lead the readers into the objective of the present work. Please check.
Materials and Methods:
- I struggled quite a lot in this section. It is really confusing all the day. I couldn't understand clearly which data was collected, from where, by who,, when,...
- First, information on the study design, protocol if published, entities clearance for study development, consents,...it should all be presented at first in this section.
- Again, abbreviations without definition at its first use (GA,...) or defined twice with different words (MI,..) , please revise.
- P. 2, l. 80 - "Oral Health Survey for England" - inconsistent with what's in P. 3, l. 99.
- Please add reason for selection of these data sources, and quick description of their objective, population studied and relevant topics.
- "Consent" - who consented, who approved the consent form used? These legal requirements have to be reported.
- Why just some of the data was collected in each source? Can't understand. Please revise and add.
- Add company, city, country in all software, equipment used.
Results:
- Since the methods section was so hard to understand, results section reading is compromised. Although, it seems that authors could achieve a better descriptive section of their results.
- Table 2 and other tables - please revise and correct lines/columns. For example, table 2 has an additional line with no results "weight status", because it was inserted into the same column as its subheadings. Please revise.
- Table 2 - Are the results for BiB, the same as total? Why having a whole column with one result and a lot of "n.a.". These table should be rewritten. Footnote does not require "*", only if "*" has a meaning. Also, in the first column, headings are not properly written because the heading is followed by data presentation without any separation or indication. Please revise.
- The term "predicted" is the one that got me most disappointed with the manuscript writing. It is not appropriate to state "predict" in a retrospective cohort study. It is not possible. Retrospective cohort lets us understand better what happened and to calculate incidence. Please revise.
Discussion:
- This section can be highly improved after all the previous ones, that will naturally confer data to discuss here.
- "being male were significant predictors of overweight/obesity" - this sentence is one example that "predict" is not at all a proper term to describe the analysis performed. Please revise.
Appendixes:
- Overall I think there's repeated data, not needed at all. Try to be concise and give the data in a way that's "reader-friendly", and that is, in fact, supplementary data for the present work. Please consider.
References
- References are not in accordance to the journal guidelines. Please revise.
Since this work is a retrospective one, authors can more easily go back to the data and find what's needed to improve the work.
Author Response
Please see attachement

Round 2
Reviewer 4 Report
I congratulate the authors for addressing my previous comments and suggestions. Now there's the need for writing review.
Thank you for your direct response to each point I've mentioned at the first round and the effort in improving your work.
Regards